# Assessment of Health-Related Quality of Life, Medication Adherence, and Prevalence of Depression in Kidney Failure Patients

**DOI:** 10.3390/ijerph192215266

**Published:** 2022-11-18

**Authors:** Muhammad Daoud Butt, Siew Chin Ong, Fatima Zahid Butt, Ahsan Sajjad, Muhammad Fawad Rasool, Imran Imran, Tanveer Ahmad, Faleh Alqahtani, Zaheer-Ud-Din Babar

**Affiliations:** 1School of Pharmaceutical Sciences, Universiti Sains Malaysia, Penang 11800, Malaysia; daoudbutt@student.usm.my; 2Department of Pharmacy, Faculty of Biological Sciences, Quaid-i-Azam University, Islamabad 15320, Pakistan; 3Lahore General Hospital, Lahore 54000, Pakistan; faateenmutriz@gmail.com (F.Z.B.); drah7an@gmail.com (A.S.); 4Department of Pharmacy Practice, Faculty of Pharmacy, Bahauddin Zakariya University, Multan 60000, Pakistan; fawadrasool@bzu.edu.pk; 5Department of Pharmacology, Faculty of Pharmacy, Bahauddin Zakariya University, Multan 60000, Pakistan; imran.ch@bzu.edu.pk; 6Institute for Advanced Biosciences (IAB), CNRS UMR5309, INSERM U1209, Grenoble Alpes University, 38400 Saint-Martin-d’Hères, France; tanveer.ahmad@univ-grenoble-alpes.fr; 7Department of Pharmacology and Toxicology, College of Pharmacy, King Saud University, Riyadh 11451, Saudi Arabia; 8Department of Pharmacy, University of Huddersfield, Huddersfield HD1 3DH, UK; z.babar@hud.ac.uk

**Keywords:** kidney failure, health-related quality of life (HRQOL), medication adherence, hemodialysis, depression

## Abstract

Background: Kidney failure is a global health problem with a worldwide mean prevalence rate of 13.4%. Kidney failure remains symptomless during most of the early stages until symptoms appear in the advanced stages. Kidney failure is associated with a decrease in health-related quality of life (HRQOL), deterioration in physical and mental health, and an increased risk of cardiovascular morbidity and mortality. This study aimed to evaluate the factors associated with decreased HRQOL and other factors affecting the overall health of patients. Another objective was to measure how medication adherence and depression could affect the overall HRQOL in patients with kidney failure. Methodology: The study used a prospective follow-up mix methodology approach with six-month follow-ups of patients. The participants included in the study population were those with chronic kidney disease grade 4 and kidney failure. Pre-validated and translated questionnaires (Kidney Disease Quality of Life-Short Form, Hamilton Depression Rating Scale Urdu Version, and Morisky Lewis Greens Adherence Scale) and assessment tools were used to collect data. Results: This study recruited 314 patients after an initial assessment based on inclusion criteria. The mean age of the study population was 54.64 ± 15.33 years. There was a 47.6% male and a 52.4% female population. Hypertension and diabetes mellitus remained the most predominant comorbid condition, affecting 64.2% and 74.6% of the population, respectively. The study suggested a significant (*p* < 0.05) deterioration in the mental health composite score with worsening laboratory variables, particularly hematological and iron studies. Demographic variables significantly impact medication adherence. HRQOL was found to be deteriorating with a significant impact on mental health compared to physical health. Conclusions: Patients on maintenance dialysis for kidney failure have a significant burden of physical and mental symptoms, depression, and low HRQOL. Given the substantial and well-known declines in physical and psychological well-being among kidney failure patients receiving hemodialysis, the findings of this research imply that these areas related to health should receive special attention in the growing and expanding population of kidney failure patients.

## 1. Introduction

Chronic kidney failure is a public health problem worldwide and can lead to conditions such as kidney failure and cardiovascular problems [1]. In different regions of the world, kidney disease has already reached an epidemic rate in people aged 20 years or above who are the most affected by this disease. The total number of men and women with kidney failure was 225.7 million and 271.8 million, respectively, by 2010 [2]. According to new research, the global prevalence of kidney failure was 9.1% in 2017 (697.5 million cases). Women and girls had a higher global majority of kidney failure than men and boys (9.5%) (7.3%) [3]. About a third of all kidney failure cases were in China (132.3 million) and India (115.1 million), with 10 countries having more than 10 million cases and 79 countries having more than 1 million cases [3,4].

Considering the high incidence of kidney failure and the challenges posed by the patients undergoing dialysis, rehabilitating one’s life is crucial. Patients suffer from different emotional and physical challenges during kidney failure [5]. Survival is not only the goal of placing the patient on a strenuous dialysis procedure but also improving the patient’s health-related quality of life (HRQOL). Understanding the deterioration of HRQOL is the most crucial aspect that can affect burden and mortality in these patients [6,7]. In patients with chronic kidney disease (CKD), including those with end-stage kidney disease (ESKD), monitoring the functional status and subjective state of well-being of a patient as they relate to a health condition, collectively known as HRQOL measurements, is of particular importance.

HRQOL has been significantly affected among patients with advanced chronic kidney disease and kidney failure. The fact that patients with CKD of various racial and ethnic backgrounds have varying HRQOL scores confirms the need to individualise the notion of HRQOL to evaluate critical aspects of life in patients and incorporate these domains into an all-encompassing plan of care. The urgency of HRQOL measurement and the need to advance our multidimensional tools with technology and a more patient-centered view of HRQOL is highlighted by these recent findings [8,9].

In dialysis patient populations, low HRQOL scores measured utilizing the Medical Outcomes Study Short Form-36 (SF-36) were associated with hospitalization and death in studies [10,11,12,13]. There were substantial HRQOL studies conducted before in patients with kidney failure. However, little has been done in Pakistan. Walters et al. assessed HRQOL at the beginning of dialysis therapy and found that in patients who began hemodialysis therapy (HD), HRQOL scores were significantly lower than in established long-term HD patients [14,15,16,17,18,19].

The Dialysis Outcomes and Practice Patterns Study (DOPPS) found a strong link between depressive symptoms and mortality and hospitalization rates [20]. On the contrary, the Choices for Healthy Outcomes in Caring for End-Stage Renal Disease (CHOICE) study found a link between high depressive symptoms and an increased risk of cardiovascular events [21]. However, this link between depression and clinical outcomes has not been thoroughly investigated in the early stages of kidney failure. A recent study by Wirkner et al. investigated an association between comorbid conditions and diabetes. The study concluded that advanced stages of CKD have significantly affected the overall HRQOL of the patients. Patients with kidney failure and undergoing dialysis were also found to have lower physical composite scores [22].

Therefore, it is crucial to assess all such parameters while dealing with a patient’s critical condition with kidney failure. This study was carried out to emphasize the use of tools to measure the HRQOL of patients and treatment outcomes [23,24,25,26]. Patient perceptions and feelings regarding their functionality and well-being are captured through patient-centered outcomes, such as HRQOL. In patients with CKD grade 4, outcome indicators are fundamental because they guide treatment objectives and strategies. However, there is little information on patient-centered outcomes in humans, especially in people who do not receive kidney replacement therapy and have CKD grade 4. More critically, there is a dearth of research in this patient population on the association between HRQOL and drug-related characteristics such as medication burden and adherence [27,28].

Previous studies have found that drug adherence rates range from 33.0 to 87.7%. Seng et al. investigated medication adherence in kidney failure. A pooled medication adherence rate of 67.4% (95% confidence interval (CI) 61.4–73.3%) was found in a meta-analysis of 54,652 patients. Between prospective and retrospective studies, the prevalence of adherence to medication among patients with dialysis kidney failure was similar (68.8%; 95% CI 61.1–76.6%) vs. (65.8%; 95% CI 57.0–74.6%) [28,29,30]. According to the study reported in Pakistan, 74% of patients with kidney failure experienced pruritus, negatively impacting their quality of life. Poor quality of life affects patients with mild to severe CKD-related pruritus. The hemodialysis patients had a lower HRQOL score as their pruritus intensity increased [31].

In Pakistan, kidney failure is the most common cause of morbidity and mortality [32]. According to the 2016 Pakistan National Kidney Federation Registry, 5935 patients were admitted to different dialysis units across the country, totaling 891 hemodialysis machines [33]. The data available have certain limitations as many large dialysis centers are not contributing their data to the registry.

There is a significant upsurge in the prevalence of kidney failure worldwide, particularly in low- to middle-income countries [34]. It is essential to evaluate the factor that could impact the quality of treatment and design a patient support program to increase patients’ involvement in the self-management of the disease condition. This study aimed to evaluate the factors associated with reduced HRQOL and other factors that affect the overall health of patients. Another objective was to measure how medication adherence and depression could impact the overall HRQOL of patients with kidney failure.

## 2. Materials and Methods

### 2.1. Study Design

This multicenter prospective cohort follow-up study was carried out in Rawalpindi and Islamabad districts. The study was conducted from April 2019 to February 2020. The patients were recruited from the Nephrology Unit of 3 major tertiary care hospitals, i.e., Federal Government Polyclinic Hospital, Islamabad, Pakistan Institute of Medical Sciences, Islamabad, and District Headquarter Hospital Rawalpindi, Pakistan.

### 2.2. Inclusion and Exclusion Criteria

A total of 400 patients were recruited for the study. After initial screening, 314 participants met the eligibility criteria for the study. All patients aged >18 years with an eGFR value of <30 mL/min/1.73 m^2^ diagnosed by a physician as a patient having CKD grade 4 and kidney failure grade 5 were enrolled in the study. All patients diagnosed with kidney failure were graded based on their history and investigation conducted in the hospital’s medical unit. According to hospital guidelines, CKD-EPI was used to estimate the eGFR; however, to reduce factors such as weight and body mass index, Cockcroft–Gault formulae were used to determine the eGFR level of the participants. The final diagnosis to determine the grades of kidney failure was made using multiple creatinine assessments. As per the guidelines provided by the study centers, they used Cockcroft–Gault formulae to give better risk stratification in cardiovascular events [35].

Participants attended a baseline clinic after being referred by their physicians. All patients were informed about the purpose of the study and written consent was obtained during the patient recruitment phase. We excluded patients diagnosed with malignancy, hospitalized patients, and patients who did not have complete laboratory investigations available at the time of recruitment. We also excluded patients with cognitive disabilities or any other mental illness who could not correctly respond to our questionnaire or other limitations that hindered their participation in this study.

### 2.3. Tools

The following tools were used to achieve the study’s goal.

### 2.4. Sociodemographic Data Sheet

It was designed by the study’s principal investigators to assess the relevant socio-demographic variables, e.g., age, gender, education status, duration of kidney failure treatment, and comorbidity condition.

### 2.5. Laboratory Investigation Assessment Datasheet

During the baseline visit, the serum potassium, calcium and phosphorus, hemoglobin, and urea levels was collected from medical records. The same laboratory investigation was also assessed after the completion of the follow-up visit at six months’ duration.

### 2.6. Assessment of Depression, HRQOL, and Medication Adherence

The study included different Urdu pre-validated and translated questionnaires to achieve the study’s objectives. All these questionnaires have been validated for use among the Pakistani population. Therefore, no further validation test was measured during the study [20,21,22,23].

### 2.7. Kidney Disease Quality of Life-Short Form (KDQOL-SF-36)

The Urdu version of the KDQOL-SF-36 is reliable and validated for assessing HRQOL in kidney disease patients on dialysis in Pakistan [36]. The KDQOL- SF-36 consists of Physical Health Composite Summary (PCS), Mental Health Composite Summary (MCS), and Kidney Disease Composite Summary (KDCS) domains. These can be measured using the HRQOL scores for physical functioning, physical disabilities, pain, emotional wellness, social functioning, and energy. The KDQOL-SF-36 has five measures, including two essential HRQOL scales (12 items) from the SF-12 version 1 and three kidney-specific scales (24 items total). In the general population, the SF-12 PCS and MCS are assessed using a T-score methodology (mean = 50, SD = 10), with higher scores indicating improved HRQOL.

### 2.8. Hamilton Depression Rating Scale Urdu Version (HAM-D-U)

The Hamilton Depression Rating Scale was used in patients who speak Urdu. The HAM-D-U is an effective tool used for determining the magnitude of depression. Internal accuracy (Cronbach alpha 0.71), test–retest reliability, and inter-rater reliability were reasonably good for the Urdu version of the HAM-D (HAM-D-U) [37]. A HAM-D-U score of 0–7 is regarded as normal (or in clinical remission), but a score of 20 or more is deemed severe (showing at least moderate severity).

HAM-D-U scores are interpreted as follows:No depression at all: <8Mild (subthreshold): 8–13Moderate (mild): 14–18Severe (moderate): 19–22Very severe (severe): >23

### 2.9. Morisky Levine Greens Adherence Scales (MLGS)

The Morisky Levine Green adherence questionnaire (MLGS) was used to evaluate medication adherence. It is a four-item questionnaire with high reliability and validity that has proven to be particularly useful in chronic conditions such as hypertension, kidney disease, and other cardiovascular diseases. It assesses deliberate and accidental adherence using criteria such as forgetfulness, carelessness, and stopping the drug while feeling better and worse. Each question in the survey was answered with a 0 or 1 response. The total score of the four elements represents the degree of prescription adherence. Medication adherence is indicated by a score of 4, while a score of less than 4 indicates non-adherence. The questionnaire was translated into Urdu using the standard forward–backwards process, yielding a Cronbach alpha of 0.570, within the appropriate range (0.45–0.9) [38,39].

### 2.10. Study Flow

The tools were administered in the following manner:

All questionnaires were administered to patients using a self-reporting approach. In order to validate the responses of the patients, researchers conducted random interviews with the patients during their dialysis procedure. The details schematic flow of patient enrollment, data collection and questionnaire administration are displayed Figure 1 and Figure 2.

### 2.11. Ethical Approval and Consent to Participate

The Bio-Ethics Committee (BEC) of Quaid-i-Azam University Islamabad permitted this prospective observational study in Tertiary Care Hospitals (BEC-FBS-QAU2017-13). The study was also approved by the Ethical Review Board of Shaheed Zulfikar Ali Bhutto Medical University (1-1/2015/ERB/SZABMU). Informed written consent was taken from all the participants included in this study. The study was conducted per the Declaration of Helsinki and reported according to “The Strengthening the Reporting of Observational Studies in Epidemiology (STROBE)” guidelines.

### 2.12. Statistical Analysis

Primary data were entered into Microsoft Excel 2010, and anonymous patient coding was used to generate secondary data. The data were analysed using the Statistical Package for Social Sciences (SPSS^®^ IBM version 20, Armonk, NY, USA) program. The KDQOL SFTM1.3 scoring program v.2.0 (Santa Monica, CA, USA) was used to compute and calculate PCS and MCS. Continuous variables were expressed in terms of means and standard deviations, whereas categorical variables were expressed in frequencies and percentages. To summarize sample characteristics, descriptive statistics were used.

Moreover, normality tests were assessed with the Shapiro–Wilk test. Data had normal distribution; therefore, continuous data’s mean and standard deviation have been presented. Categorical variables were presented as numbers and percentages. The Mann–Whitney U test was used to compare the depression and HRQOL domains. A *p*-value of <0.05 was considered statistically significant, and a *p*-value of < 0.01 was considered highly significant. The multivariate analysis was also performed using Bonferroni Correction. Binary logistic regression was used to examine factors related to medication non-adherence and depression, and effect sizes were reported using odds ratios (ORs) and 95% confidence intervals (CIs). The study variable includes dependent variables (KDQOL Domain and Scale, MCS, PCS, medication adherence, and depression) and independent variables (sociodemographic parameters, laboratory parameters).

## 3. Result

### 3.1. Patient Characteristics

During the study recruitment period, 400 patients were enrolled at the treatment sites. Eighty-six patients did not meet the eligibility criteria and were excluded. In total, 314 patients participated in the study and completed KDQOL, HAM-D-U, and MLGS. Figure 2 shows the schematic flow of the study. Of the total 314 patients, 39.8% (*n* = 125) were CKD grade 4, 9.2% (*n* = 29) with grade 5 not on hemodialysis yet, and 50.9% (*n* = 160) were at kidney failure grade 5 on hemodialysis.

The demographic characteristics of the study participants are shown in Table 1. The mean age of the participants was 54.64 ± 15.33. The majority of the survey participants were women, with a significant proportion of the study population from the age group 41–60 years. The number of people with BMI in the normal range was 53.0%. The study population selected through convenience sampling comprised different levels of disease, treatment approaches, and durations.

The comorbidity conditions among the study populations are depicted in Figure 3. Among the population with CKD, diabetes and hypertension were the most prominent underlying comorbid conditions, i.e., 62% and 55%, respectively. Among the study population, 19% had various etiological factors, e.g., smoking, and illicit drug use. Most patients had a family history of kidney disease and a strong history of heart disease. A large group of people also had unknown etiology and represented 30% of the study population.

### 3.2. Laboratory Parameters

There were abnormalities in laboratory parameters, and their impact was also analysed in the overall HRQOL of patients. Statistical analysis suggested that hematological and serum iron studies were laboratory parameters that significantly affected the mental composite score (*p*-Value < 0.05). Contrary to this, the physical composite score was greatly affected by parameters such as hematological serum iron and renal function concentration in the blood, as shown in Table 2.

### 3.3. Depression

In the study, the 21-item Hamilton Depression Rating Scale (HAM-D-U) was used to determine the study population’s pre-study and post-study analysis of depression levels.

It was observed that throughout treatment, there was no educational or counselling support provided to any of the patients. Analysis of the depression data was given by 314 patients in the pre-assessment questionnaire and 244 in the post-assessment questionnaire. A total of 45 patients refused to fill out the post-study HAM-D-U.

The frequency distribution of the depression assessment at baseline found that there was a significantly high depression level in both kidney failure populations (Figure 4). There were only a few cases with a normal-to-moderate level of depression.

Upon the follow-up visit, it was observed that there was a significant increase in depression levels among the study population. Respondents indicated a higher level of depression in both the study populations compared to their previous state. However, further analysis with the Pearson Chi-square test found no significant difference between the baseline and post-study depression levels (*p*-value > 0.05).

### 3.4. Medication Adherence

Around 38.8% (*n* = 122) of the 314 participants scored 4 and had high adherence, while the majority (61.2%, *n* = 192) belonged to a non-adherent population with a score of less than 4. In addition, the findings revealed no statistically significant relationship between adherence and gender, age, marital status, education level, smoking status or the number of medications (*p* > 0.05).

### 3.5. HRQOL

A total of 314 patients completed the KDQOL-SF-36 questionnaire during a baseline visit. As shown in Table 3, even though patients with grade 4 had a lower level of the physical score as compared to those having grade 5, there were no significant differences seen in the SF-36 PCS scale (31.28 ± 7.91 vs. 36.28 ± 8.41, *p* = 0.68). However, the overall mental health composite measured by the MCS at baseline was significantly lower for the grade 4 population than the grade 5 population, i.e., 36.66 ± 6.57 and 48.66 ± 5.44, respectively (*p*-value < 0.05).

For the KDQOL-SF-36 domain, the Mental Health Composite was the factor that affected the study population the most, which showed a statistically significant level in both the study groups (*p*-value < 0.05). The other major KDQOL-SF-36 domains that impacted both populations were Physical Functioning, Role Limitations-Physical, and Emotional well-being, as well as energy and fatigue evident from Table 3.

In the follow-up visit, 244 patients agreed to respond to the questionnaires. In the post-study, 35 patients with grade 4 and 10 patients with grade 5 did not respond. In the post-analysis, a marked deterioration was seen in KDQOL-SF-36 domains and scale parameters among patients with advanced kidney failure. Grade 4 patients scored significantly lower than grade 5 in the Problem’s list/Symptom, Burden of Kidney Disease, and Mental Health composite domains (*p* < 0.05). In the KDQOL-SF-36 scales, there was significant deterioration in Physical function, Emotional well-being, and Energy level in patients with grade 5 (*p*-value < 0.05).

### 3.6. Medication Adherence and HRQOL

The PCS of the HRQOL improved over time in adherent individuals and declined in non-adherent individuals. Additionally, after controlling for age, gender, and baseline eGFR, medication non-adherence demonstrated a significant negative correlation with a change in the PCS of the HRQOL, as evident from Table 4. After adjusting for the same covariates, there was no significant correlation between medication non-adherence and a change in the MCS of the HRQOL.

### 3.7. Depression and HRQOL

Depressive symptoms were inversely correlated with PCS and MCS scores in the unadjusted longitudinal analysis. Depressive symptoms were independently reduced following multivariable adjustment and negatively correlated with the PCS and MCS scores, as shown in Table 5. When depressive symptoms were considered a dichotomous variable in unadjusted analyses, moderate-to-severe depression symptoms presented negative correlations between the PCS and MCS. In multivariable analyses, the PCS and MCS scores were independently and negatively related to the prevalence of moderate-to-severe depressive symptoms.

## 4. Discussion

The study was conducted in two major cities in Pakistan and is the first-ever follow-up study investigating the point of concern for patients with kidney failure in grade 5 and CKD grade 4. According to the results of this study, targeted patients with kidney failure grade 5 who are not on chronic renal replacement therapy have a similar overall burden of symptoms and depression and a poor HRQOL. The results derived from this research have significant clinical implications for both patients and care providers.

Despite studies showing that patients with kidney failure grade 4 have poor physical and psychosocial health, the clinical, medication, and patient-related factors that cause symptoms, depression, and poor HRQOL in the targeted patient population are unknown. Patients tend to develop symptoms, depression, and poor HRQOL if they lose much weight without losing kidney function. Therefore, it is critical to determine whether this is due to metabolic disturbances, retained uremic contaminants, comorbid medical conditions, anxiety about kidney failure, the possible need for renal replacement therapy in the future or other causes to administer adequate care. Due to the use of different tools for detailed analysis, the results derived in this study have significant implications for kidney failure patients.

Nonadherence to medication did not affect HRQOL at baseline but caused a gradual reduction in physical HRQOL (SF36-PCS) scores, although the mental HRQOL of all individuals decreased significantly during the follow-up period. Studies have shown that baseline eGFR levels influence physical deterioration in HRQOL [40]. However, we could not discover an association between initial eGFR and alterations in HRQOL over time in the current study.

It was also observed that the physical and mental health composite tends to deteriorate in patients with kidney failure. Previous studies reported that patients with moderate-to-progressing kidney disease had a lower PCS than MCS [27]. A previous study on kidney failure grade 1–4 patients reported that the PCS and MCS scores were 32.1 ± 8.1 and 40.6 ± 11.1, respectively. In the current study, the severity of the disease, particularly in individuals with kidney failure, had a considerable impact on HRQOL. This link has also been discovered in other investigations where HRQOL scores are compromised in patients with intermediate kidney failure and deteriorate over time in dialysis patients [9,41].

A study by Ajeebi and his colleagues showed that patients with advanced kidney failure could likely be unaware of the effects of chronic renal replacement therapy on their physical and mental health [42]. Following this school of thought, it is evident that kidney failure patients need chronic dialysis for a drastic lifestyle change [43]. Based on the current study findings, there is evidence that patients with kidney failure experienced a significant difference in their physical and psychosocial well-being during the transition period of the disease.

However, in the case of patients with kidney failure, complaints of sleep problems, muscle cramps, dry mouth, and lightheadedness are the most frequent. Although these discrepancies did not reach statistical significance after several comparisons, the biologi-cal plausibility of such differences warrants further investigation. In light of the results, it is evident that patients with kidney failure have lower scores on the SF-36 physical func-tion subscale. However, there were no variations in PCS scores due to this observation. As per the study referred to, this provides preliminary information on the subdomains of HRQOL that may differ between these two populations [24]. In a study conducted on the patients undergoing kidney transplantation and those on hemodialysis or peritoneal dialysis, the findings provided evidence that there was a significant improvement in HRQOL after one year of kidney replacement and dialysis [44]. The comparison between the current results and the existing literature provides evidence that patients with kidney failure are equally treated for health issues, including anemia, depression, sleeplessness, and bone and muscle diseases, that lead to the deterioration of their quality of life. Therefore, in the context of grade 5 kidney failure, patients should be informed about the unpredictability of their disease as it is difficult to anticipate the disease trajectory based on known predictive factors of mortality and the advantages and disadvantages of dialysis treatments. According to a study by Nataatmadja et al. (2021), of 3000 patients with an average age of 73 years, 60% regretted starting dialysis rather than having opted for conservative treatment [4]. As far as possible, the decision to initiate dialysis in patients with grade 5 kidney failure must be made personally between the patient, family members, the attending physician, and the referring nephrologist.

Numerous studies have shown that dialysis has a detrimental effect on depression and that patients who experience severe depression are more likely to die. According to 15 large-scale investigations, there is an important link between depression and mortality among dialysis patients. Several longitudinal studies that evaluated the recurrent measurement of depression found that patients receiving dialysis therapy had significantly greater mortality risks when depressive symptoms were present. According to studies, early dialysis therapy initiation is associated with depression [45,46,47].

The general demographic features of our sample of grade 4 kidney failure were close to those of the United States population of grade 4 chronic kidney disease. On the contrary, the study’s kidney failure cohort had a higher proportion of men than the total population of kidney failure patients. The result supported the study mentioned above, as it was discovered that the depression rate was observed to worsen with time; a linear increase was identified from the start of the study until the second administration of the questionnaire 6 months later. Possible causes for this finding are likely to be lifelong dialysis therapy with at least 3 dialysis operations per week, patients taking too many medications at once, the economic strain on patients and their families, and changed familial and social ties. In a group of chronic HD patients, depression symptoms increased linearly and a link was discovered between poor sleep quality, unemployment, pruritus, hypoalbuminemia, diabetes, and depressive symptoms [48].

The findings of the current study and the prior research provided evidence that patients suffering from kidney failure are likely to develop mental health issues that the prolonged effects of illness could cause.

Chronic renal diseases are frequent in the population but often remain challenging to demonstrate clinically, as their symptoms are crude [49]. It is thus not uncommon for patients to develop end-stage renal failure, having presented very few symptoms or only vague and not very specific symptoms (fatigue, inappetence, etc.). In extensive population studies, it was found that only 23% of people with kidney failure were aware of their diagnosis [50]. According to the survey by Live and Zhang (2019), kidney failure is, in most cases, secondary to hypertension, diabetes, immunological disease, etc. Nevertheless, there are several kidney diseases that, although rare, could be diagnosed early, thanks to typical clinical and biological signs (red flags). However, even these diseases are often underdiagnosed, further prolonging the diagnostic error of these patients and their early management [51,52,53].

In addition, the treatment of hemodialysis patients includes all possible measures to reduce the impact of symptoms encountered in kidney failure [48]. These symptoms are pain, pruritus, nausea, headache, fatigue, anxiety, and depression. A total of 80% of patients suffered from chronic kidney pain and went to dialysis due to underlying mental health complications, especially depression. It has been stated in the study of Duane et al. that diseases affecting the kidneys are frequent but remain asymptomatic for a long time, leading to a diagnostic delay with consequences for the management of the patient [53]. However, convincing warning signs, whether cutaneous, articular or related to electrolyte abnormalities, should put the chip in the ear and guide the practitioner to the renal origin of the pathology. Therefore, certain rare diseases affecting the kidneys are discussed here based on clinical vignettes, emphasizing the biological and clinical signs that lead to their diagnosis [54].

Moreover, mental health depends on individual, genetic, and hormonal factors, but also relational, community, and societal factors. Patients of kidney failure of all ages suffer from certain mental disorders more frequently, with more comorbidities. It also impacts their physical health and family and social balance. Specificities exist at the clinical level, as well as concerning the use of psychotropic drugs, their side effects, and adherence to care. The primary care physician’s role is essential for preventing and detecting mental disorders in kidney failure patients for their psychological support, guidance, and follow-up [41,55].

The current study has some limitations. The study only has a 6-month follow-up period; the findings should be interpreted cautiously. Although the variables have been adjusted for changes in physical activity, the possibility of residual confounders cannot be ruled out as one of the typical limitations of all cohort studies. For example, the health consciousness of an individual eating healthier also makes other more beneficial lifestyle changes that the questionnaire could not completely capture. A multicenter investigation with a large sample size and an extended follow-up period is required to corroborate the findings of the current study. Furthermore, future research in this domain needs to consider the comparison of mentioned health-related domains in a much more comprehensive sample of patients with kidney failure, including those with severe comorbid illnesses. In this study, which investigated several variables associated with low HRQOL in patients with kidney failure, depression significantly predicted a lower HRQOL. Therefore, our study emphasizes the potential need for randomized controlled trials in future research to determine the temporal relationship between low HRQOL and depression in kidney failure patients and to further assess the impact of early identification of depression and appropriate treatment of depression on the health outcomes of these patients. Despite these drawbacks, our study’s strength is that we performed thorough analyses to look at factors associated with HRQOL in patients with kidney failure, including medication adherence, depression, and modifiable and nonmodifiable clinical features.

## 5. Conclusions

The patients on maintenance dialysis for kidney failure have significant burdens of physical and mental symptoms, depression, and low HRQOL. Given the substantial and well-known declines in physical and psychological well-being among patients with kidney failure receiving hemodialysis, findings drawn in this research imply that these health-related areas should receive special attention in the vast and expanding population of patients with kidney failure.

## Figures and Tables

**Figure 1 ijerph-19-15266-f001:**
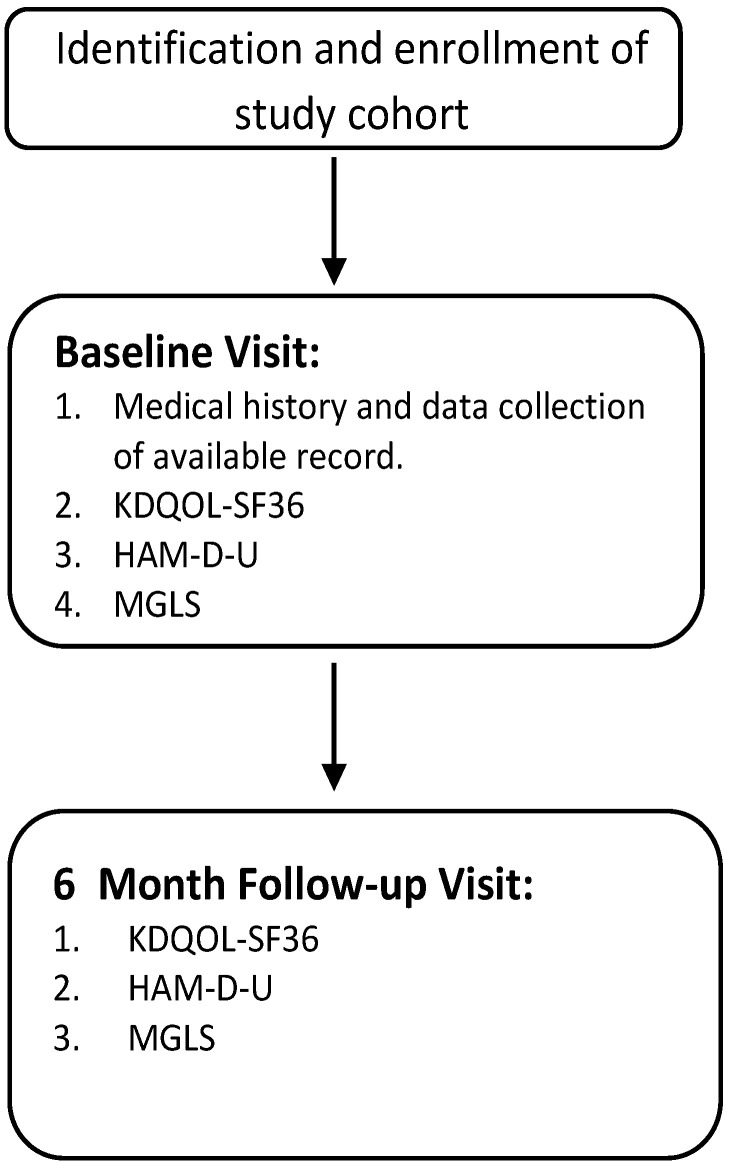
Schematic representation of data collection and questionnaire administration.

**Figure 2 ijerph-19-15266-f002:**
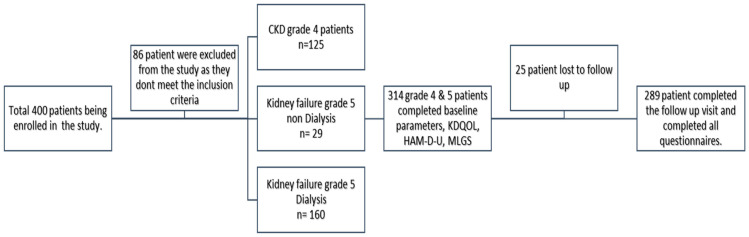
Schematic representation of patient enrollment and data collection.

**Figure 3 ijerph-19-15266-f003:**
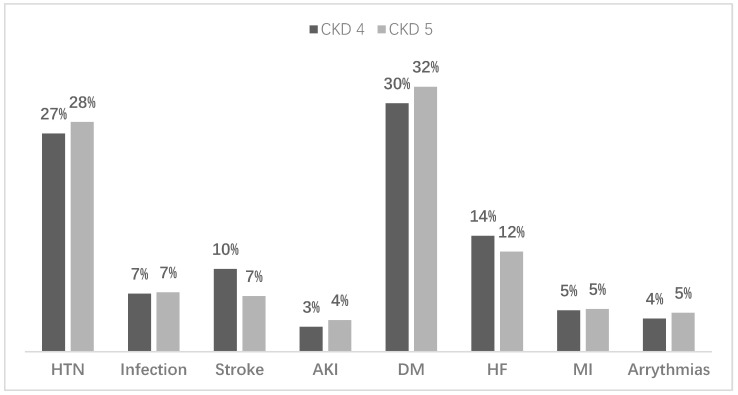
Comorbidity conditions among the study groups. CKD4: Chronic kidney disease grade 4 and grade 5. HTN: Hypertension, AKI: Acute Kidney Injury, DM: Diabetes Mellitus, HF: Heart Failure, MI: Myocardial Infarction.

**Figure 4 ijerph-19-15266-f004:**
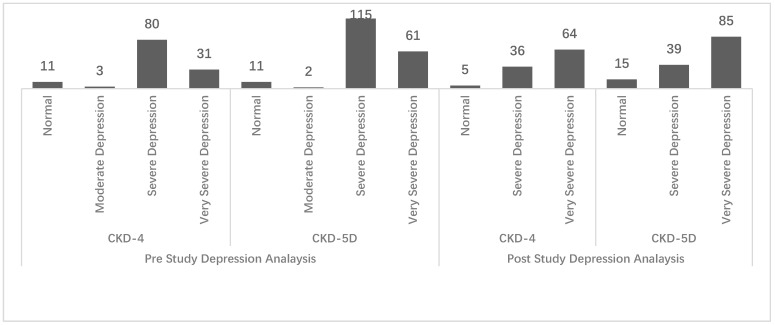
Depression level distribution (frequency) among study groups pre- and post-study.

**Table 1 ijerph-19-15266-t001:** Demographic variables Clinical Characteristics (*n* = 314).

Parameters		*n* (%)
**Gender**	Female	164 (52.4)
	Male	150 (47.6)
	Age mean (±SD)	54.64 (±15.33)
**Age group (years)**		
	<40	57 (18.2)
	41–60	145 (46.2)
	>60	112 (35.7)
	BMI mean (±SD)	20.08 (±3.65)
**BMI * Classification**		
	Underweight	102 (32.4)
	Normal	167 (53.0)
	Overweight	32 (10.2)
	Obese	12 (3.8)
**Socioeconomic status ****		
	Low	94 (29.9)
	Middle	197 (62.8)
	High	23 (7.3)
**Education Level**		
	Uneducated	185 (59.9)
	Educated	129 (41.08)
**Marital Status**		
	Single	32 (10.1)
	Married	282 (89.9)
**Smoking Status**		
	Current Smoker	73 (23.2)
	Non-Smoker	241 (76.8)
**Employment**		
	Unemployed	120 (54.5)
	Employed	100 (45.5)
**Grade of kidney failure**		
	Chronic kidney disease grade 4	125 (39.8)
	Kidney failure grade 5 (Non-Dialysis)	29 (9.2)
	Kidney failure grade 5 (Hemodialysis)	160 (50.9)
**Kidney failure Duration**		
	1 year	91 (29)
	2–4 years	153 (48.7)
	>5 years	70 (22.3)

* BMI: Body Mass Index. ** The socioeconomic status was assessed using the World Bank database’s definition to determine the economic situation. *Low Income **≤**
**USD 1537.9***, *Middle Income **USD 1537.9–2581.9***, and *High Income **≥ USD 2581.9***.

**Table 2 ijerph-19-15266-t002:** Multivariate analysis of laboratory parameters and their correlation with Quality of Life.

Biochemical Parameters **	Normal Range	Grade 4 Pre Laboratory Parameters	Grade 4 Post Laboratory Parameters	PCS *p*-Value	MCS *p*-Value	Grade 5 Pre Laboratory Parameters	Grade 5 Post Laboratory Parameters	PCS *p*-Value	MCS *p*-Value
		Mean ± SD	Mean ± SD			Mean ± SD	Mean ± SD		
**Hematology**	Hemoglobin (11–18 g/dL)	8.9 ± 1.61	8.8 ± 1.7	**<0.05 ***	0.471	9.95 ± 1.69	8.85 ± 1.62	**<0.05 ***	0.412
**Serum Iron Studies**	Ferritin (250–450 µg/dL)	147.60 ± 81.52	136.59 ± 82.0	**<0.05 ***	**<0.05 ***	147.60 ± 81.52	136.59 ± 82.0	**<0.05 ***	**<0.05 ***
	Iron (50–170 µg/dL)	175.1 ± 31.38	164.26 ± 31.56	**<0.05 ***	**<0.05 ***	176.4 ± 31.38	174.16 ± 32.8	0.163	**<0.05 ***
**Renal Function Test**	Creatinine (0.5–1.1 mg/dL)	10.9 ± 3.2	14.6 ± 2.71	**<0.05 ***	0.241	14.46 ± 9.1	10.66 ± 3.01	N/A	N/A
	Urea (13–43 mg/dL)	174.03 ± 57.78	163.68 ± 72.13	**<0.05 ***	**<0.05 ***	160.3 ± 67.8	154.3 ± 62.7	N/A	N/A
**Liver Function Test**	Bilirubin (0.3–1.2 mg/dL)	1.11 ± 0.31	1.10 ± 0.32	0.277	0.432	1.31 ± 0.41	1.10 ± 0.22	0.289	0.093
**Serum Electrolytes**	Calcium (8.4–10.2 mg/dL)	6.6 ± 1.2	6.7 ± 1.31	0.240	0.331	8.77 ± 1.19	8.70 ± 1.28	0.683	0.569
	Sodium (136–146 mEq/L)	137.1 ± 13.9	137.0 ± 13.0	0.760	0.491	138.26 ± 13.05	139.5 ± 13.27	0.852	0.311
	Potassium (3.5–5.1 mEq/L)	4.49 ± 0.52	4.15 ± 0.33	0.778	0.922	4.59 ± 0.42	4.65 ± 0.43	0.201	0.542
	Phosphate (35–105 U/L)	76.2 ± 14.73	73.04 ± 16.6	0.807	0.547	81.2 ± 12.11	79.04 ± 15.1	0.838	0.368

Abbreviations: PCS, Physical Composite Summary; MCS, Mental Composite Summary; g/dL, grams per deciliters; mg/dL, milligrams per deciliters; *mEq/L,* milliequivalents per liter; U/L, unit per liter; the *p*-value was calculated using the Mann–Whitney U test (non-parametric test). * Indicates the statistically significant *p*-value (<0.05). ** Pre-Laboratory parameters were collected from the patient’s medical record before the study, and post-Laboratory parameters were collected after six months from the patient’s medical record file.

**Table 3 ijerph-19-15266-t003:** Pre and Post mean assessment score of KDQOL-SF-36 Domain and Scale Component.

KDQOL Domains	No. of Items in a Scale	Mean			Mean			KDQOL Scale	No. of Items in Scale	Mean			Mean		
		Pre-Study		*p*-value	Post-Study		*p*-Value			Pre-Study		*p*-Value	Post Study		*p*-Value
		Grade 4	Grade 5		Grade 4	Grade 5				Grade 4	Grade 5		Grade 4	Grade 5	
** *Symptom/problem list* **	12	57.70 ± 11.44	63.70 ± 12.39	0.08	58.70 ± 16.63	68.70 ± 11.13	**0.002 ***	** *Physical functioning* **	10	41.68 ± 27.17	61.24 ± 8.72	**0.008 ***	40.12 ± 24.66	57.44 ± 9.12	**0.03 ***
								** *Role limitations--physical* **	4	33.31 ± 17.85	41.02 ± 11.12	**0.02 ***	33.12 ± 19.05	40.82 ± 10.72	0.28
** *Effects of kidney disease* **	8	56.45 ± 11.23	55.85 ± 10.80	0.1	57.45 ± 13.41	49.25 ± 9.10	0.35	** *Pain* **	2	13.60 ± 6.57	12.67 ± 3.41	0.41	14.11 ± 5.07	17.07 ± 4.88	0.41
								** *General Health* **	5	31.84 ± 10.35	31.11 ± 8.43	0.88	30.04 ± 11.75	34.31 ± 8.69	0.68
** *Burden of kidney disease* **	4	29.06 ± 11.59	34.06 ± 9.83	0.24	26.06 ± 11.59	34.86 ± 10.11	**0.04 ***	** *Emotional well-being* **	5	39.30 ± 9.07	51.44 ± 7.32	**0.009 ***	31.03 ± 12.17	48.42 ± 6.12	**0.000 ***
** *Physical Health Composite* **	12	31.28 ± 7.91	36.28 ± 8.41	0.68	33.81 ± 7.14	37.22 ± 8.11	0.68	** *Role limitations-- emotional* **	3	28.42 ± 11.59	28.22 ± 8.47	0.78	28.11 ± 9.11	29.62 ± 7.97	0.78
								** *Social function* **	2	18.13 ± 5.39	21.85 ± 7.77	0.45	19.03 ± 5.11	20.65 ± 7.01	0.61
** *Mental Health Composite* **	12	36.66 ± 6.57	48.66 ± 5.44	**0.009 ***	37.99 ± 6.57	47.16 ± 7.44	**0.01 ***	** *Energy/fatigue* **	4	48.50 ± 9.11	59.21 ± 5.51	**0.046 ***	47.11 ± 8.61	51.59 ± 7.91	**0.046 ***

*p*-value has been calculated using the Mann–Whitney U test (non-parametric test). * indicates the statistically significant *p*-value (<0.05).

**Table 4 ijerph-19-15266-t004:** Relationship between changes in the physical and mental component of HRQOL and medication non-adherence (MGLS).

	Outcome	Depression Defined Using MGLS as a Categorical Variable	
		Unadjusted Association	Adjusted Association
Grade 4	PCS	−3.99 (−8.29, 0.31) **	−4.64 (−9.10, −0.17) **
	MCS	1.82 (−3.12, 6.78) **	2.03 (−2.99, 7.05) *
Grade 5	PCS	−3.0(−9.9, 3.9) **	−3.4(−9.1, 0.90) **
	MCS	1.43 (−4.99, 7.85) *	1.15(−5.3,7.6) *

Data denote β coefficient (95% CI). * *p* < 0.05; ** *p* < 0.01.

**Table 5 ijerph-19-15266-t005:** Relationship between changes in physical and mental health-related quality of life health and depression (HAM-D).

	Outcome	Depression Defined Using HAM-D as a Continuous Variable		Depression Defined Using HAM-D as a Categorical Variable	
		Unadjusted Association	Adjusted Association	Unadjusted Association	Adjusted Association
Grade 4	PCS	−1.62 (−2.38, −0.86)	−1.29 (−1.87, −0.72)	−2.13 (−3.66, −0.60) *	−1.73 (−3.09, −0.37) **
	MCS	−5.32 (−6.11, −4.53)	−4.52 (−5.15, −3.89)	−9.24 (−11.27, −7.21)	−8.13 (−9.83, −6.44) *
Grade 5	PCS	−1.96 (−3.01, −0.91)	−1.68 (−2.41, −0.96)	−2.01 (−3.06, −0.97)	−1.71 (−2.87, −0.55) *
	MCS	−5.30 (−6.84, −3.77)	−4.71 (−5.98, −3.44)	−9.06 (−10.51, −7.61) **	−7.81 (−9.30, −6.33)

Data denote β coefficient (95% CI). * *p* < 0.01, ** *p* < 0.05.

## Data Availability

The authors own the data used in this study. Any request for data access should be directed to the associated author. The dataset generated during the current study was not available publicly. However, if there is any reasonable request, it can be made available from the corresponding author.

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
