# Peer review of "Assessment of Health-Related Quality of Life, Medication Adherence, and Prevalence of Depression in Kidney Failure Patients"

_ijerph, 2022, doi:10.3390/ijerph192215266_

Round 1
Reviewer 1 Report
The manuscript is well-presented. However, there are a few minor changes that need to be addressed to proceed with the publication.
1. The acronym HRQOL is not defined in the first place in the introduction.
2. The authors have failed to provide the literature evidence for the below claims within the introduction section.
“The Dialysis Outcomes and Practice Patterns Study (DOPPS) found a strong link be-tween depressive symptoms and mortality and hospitalization rates. In contrast, the Choices for Healthy Outcomes in Caring for End-Stage Renal Disease (CHOICE) Study found a link between high depressive symptoms and an increased risk of cardiovascular events. However, this link between depression and clinical outcomes has not been thor-oughly investigated in the early stages of kidney failure.”
3. In the below claim, the reference provided is wrong. Instead [16], its [13]. Revise carefully.
“Walters et al. assessed HRQOL at the initiation of dialysis therapy and found in patients beginning hemodialysis (HD) therapy that HRQOL scores were significantly lower than in estab-lished long-term HD patients.”
4. Which study authors are referring to actually? I couldn’t find any in the preceding sentence.
This study was conducted to emphasize the use of tools to measure the HRQoL of patients and treatment outcomes[19-22].
5. The authors have provided literature evidence of prior studies and have discussed the findings of others (more than one work), however, only one reference is given. Please revise carefully.
Previous studies have found that drug adherence rates range from 33.0 to 87.7%. Seng, Jun Jie Benjamin, et al. investigated medication adherence in kidney failure. A pooled medication adherence rate of 67.4% (95% confidence interval (CI) 61.4–73.3%) was found in a meta-analysis of 54,652 patients. Between prospective and retrospective studies, the prevalence of medication adherence among pre-dialysis kidney failure patients was sim-ilar [(68.8%; 95% CI 61.1–76.6%) vs. (65.8%; 95% CI 57.0–74.6% )[23]
6. The authors have conducted the proposed study in the Pakistani setting. There must be some literature background from the Pakistani population. The authors fail to provide sufficient literature background to build their arguments.
7. What was the medium of filling the questionnaires from the recruited patients?
8. In the heading of table 2, please mention whether the univariate analysis or … ? Also, define the acronyms used in the table.
9. Discussion is not sufficiently corroborated by the recent literature. The authors need to revise it to strengthen their study findings.
10. There are plenty of grammatical errors and sentence structuring issues that need to be fixed.
11. References also require careful revision.
Author Response
- The acronym HRQOL is not defined in the first place in the introduction.
Response:
Thank you for your comment. Highlighted acronym HRQOL has been defined at the first occurrence.
- The authors have failed to provide the literature evidence for the below claims within the introduction section.
“The Dialysis Outcomes and Practice Patterns Study (DOPPS) found a strong link between depressive symptoms and mortality and hospitalization rates. In contrast, the Choices for Healthy Outcomes in Caring for End-Stage Renal Disease (CHOICE) Study found a link between high depressive symptoms and an increased risk of cardiovascular events. However, this link between depression and clinical outcomes has not been thoroughly investigated in the early stages of kidney failure.”
Response:
Thank you for pointing out the missed citation. The updated citation contains the literature evidence of the whole claim in the revised introduction section. Refer to lines 83-88.
- In the below claim, the reference provided is wrong. Instead [16], its [13]. Revise carefully.
“Walters et al. assessed HRQOL at the initiation of dialysis therapy and found in patients beginning hemodialysis (HD) therapy that HRQOL scores were significantly lower than in estab-lished long-term HD patients.”
Response:
Thank you for highlighting the mistake. The references have been revised. Refer to lines 80-82.
- Which study authors are referring to actually? I couldn’t find any in the preceding sentence.
This study was conducted to emphasize the use of tools to measure the HRQoL of patients and treatment outcomes[19-22].
Response:
Thank you for highlighting the concern. The statement has been updated and the complete detail is being added to the manuscript. Refer to lines 93-99.
- The authors have provided literature evidence of prior studies and have discussed the findings of others (more than one work), however, only one reference is given. Please revise carefully.
Previous studies have found that drug adherence rates range from 33.0 to 87.7%. Seng, Jun Jie Benjamin, et al. investigated medication adherence in kidney failure. A pooled medication adherence rate of 67.4% (95% confidence interval (CI) 61.4–73.3%) was found in a meta-analysis of 54,652 patients. Between prospective and retrospective studies, the prevalence of medication adherence among pre-dialysis kidney failure patients was sim-ilar [(68.8%; 95% CI 61.1–76.6%) vs. (65.8%; 95% CI 57.0–74.6% )[23]
Reponse:
Thank you for highlighting the concern. There was an automation issue with the Endnote use. However, the updated references have been updated to the manuscript. Refer to lines 104-110.
- The authors have conducted the proposed study in the Pakistani setting. There must be some literature background from the Pakistani population. The authors fail to provide sufficient literature background to build their arguments.
Response:
Thank you for highlighting the negligence. That was missed in the 1st part; however, a complete detail has been added in the revised manuscript from the perspective of Pakistan. Refer to lines 110-113
- What was the medium of filling the questionnaires from the recruited patients?
Response:
Thank you for your comment, It is to share that medium of filling here is the self-reporting method, and afterwards, a brief interview was also conducted by the researcher to validate the response. A statement has been added in the manuscript. Refer to lines 200-201
- In the heading of table 2, please mention whether the univariate analysis or … ? Also, define the acronyms used in the table.
Response:
Thank you for your comment. The multivariate analysis was done, as mentioned in the heading of table 2. The acronyms have been defined in the revised table.
- Discussion is not sufficiently corroborated by the recent literature. The authors need to revise it to strengthen their study findings.
Response:
Thank you for the comment. The discussion section is thoroughly revised, and study findings were further corroborated with the recent literature. Refer to lines 361-422
- There are plenty of grammatical errors and sentence structuring issues that need to be fixed.
Response:
Thank you for highlighting this. The revised manuscript has been checked for grammatical errors and proofread by an expert.
- References also require careful revision.
Response:
The references have been thoroughly revised and updated in the revised manuscript.
Reviewer 2 Report
1. “Kidney failure is a public health problem globally and can lead to conditions including kidney failure and cardiovascular problems.”
This sentence needs to be corrected.
2. “Kidney failure”
The term “kidney failure” should be replaced by chronic kidney disease (CKD)
3. “HRQOL has significantly impaired both incidents and prevalent dialysis patients' grades G4 & G5.”
The sentence is unclear. Besides, patients in stage G4 of CKD should not receive renal replacement therapy as hemodialysis.
4. „The study was conducted from June 2019 to February 2020, and patient inclusion and recruitment were completed in May 2019.”
How could be the recruitment completed before the study? Recruitment means beginning of the study.
5. „19% of the population had various etiological factors, e.g., proteinuria, cigarette smoking, chronic anemia, and illicit drug use.”
Proteinuria is rather common for patients suffering from CKD. First of all, as the manifestation of both primary and secondary glomerular diseases. The same problem is with chronic anemia, which is characteristic for advanced CKD.
Author Response
- “Kidney failure is a public health problem globally and can lead to conditions including kidney failure and cardiovascular problems.”
This sentence needs to be corrected.
Response:
Thank you for your comment. The statement has been edited and corrected in the revised manuscript.
- “Kidney failure”
The term “kidney failure” should be replaced by chronic kidney disease (CKD).
Response:
Thank you for the suggestion. Currently, the terminology of CKD has been revised, enabling researchers to compare results across studies conveniently. For example, end-stage kidney failure or end-stage kidney disease, has now been termed as “kidney failure” based on the latest guideline.
Levey et al. (2020). Nomenclature for kidney function and disease: Executive summary and glossary from a Kidney Disease: Improving Global Outcomes (KDIGO) consensus conference. Journal of Nephrology, 33(4), 639–648. doi:10.1007/s40620-020-00773-6
Therefore the terminology of “kidney failure” has been used in the current manuscript.
- “HRQOL has significantly impaired both incidents and prevalent dialysis patients' grades G4 & G5.”
The sentence is unclear. Besides, patients in stage G4 of CKD should not receive renal replacement therapy as hemodialysis.
Response:
Thank you for your comment. We have amended the statement in the revised manuscript with clarification provided. Refer to lines 93-101.
- „The study was conducted from June 2019 to February 2020, and patient inclusion and recruitment were completed in May 2019.”
How could be the recruitment completed before the study? Recruitment means beginning of the study.
Response:
Thank you for highlighting the typographical error. Amendment has been made to the statement in the revised manuscript. Kindly refer to the line 126-131 in methodology section.
- „19% of the population had various etiological factors, e.g., proteinuria, cigarette smoking, chronic anemia, and illicit drug use.”
Proteinuria is rather common for patients suffering from CKD. First of all, as the manifestation of both primary and secondary glomerular diseases. The same problem is with chronic anemia, which is characteristic for advanced CKD.
Response:
Thank you for your valuable comment. The statement has been revised in the revised manuscript. Kindly refer to the updated statement in lines 251-257.
Reviewer 3 Report
By typing “quality of life and chronic renal disease” in the PubMed search string, it is possible to obtain more than 8,000 articles, therefore discussion of quality of life in chronic kidney disease patients is not a novelty. Daoud Butt et al performed a prospective study evaluating quality of life using the kidney disease quality of life short form, depression using the Hamilton depression rating scale and adherence to treatment using the Morisky Lewis Greens adherence scale. Besides, the authors collected data about comorbidity, laboratory parameters and sociodemographic ones. However, I do not think that in the present form the paper can be accepted for publication because of the following main reasons:
1) Introduction: in my opinion, it should be rewritten taking into consideration the aim of the study that it is not clearly stated.
2) Inclusion criteria: it is not clear the reason for using Cockcroft-Gault formula instead of CKD-EPI one. However, inclusion and exclusion criteria should be clearly stated.
3) Statistical analysis: it is not clear the reason why it is not stated the comparison of data collected at the beginning and at the end of the period study (if authors did it). Dependent and independent variables used in the logistic regression analysis should be reported.
4) Results: results are not clearly reported.
5) Discussion: some paragraphs are not related to results.
6) References: some references on the item the authors want to discuss are missing.
I suggest authors evaluate a recent paper published in the Journal of Clinical Medicine entitled “Comorbid depression and diabetes are associated with impaired health-related quality of life in chronic kidney disease patients (J Clin Med 2022,11,4671).
Author Response
1) Introduction: in my opinion, it should be rewritten taking into consideration the aim of the study that is not clearly stated.
Response:
Thank you for your valuable comment. The introduction section has been revised considerably with updated references to clearly state the study's aim. The section has also been revised based on the other reviewers’ comments.
2) Inclusion criteria: it is not clear the reason for using Cockcroft-Gault formula instead of CKD-EPI one. However, inclusion and exclusion criteria should be clearly stated.
Response:
Thank you for highlighting the point. Besides Cockcroft-Gault formula, the CKD-EPI was also used. . The manuscript methodology section has been revised further to elaborate on the point. Kindly refer to lines 132-149
3) Statistical analysis: it is not clear reason why it is not stated the comparison of data collected at the beginning and at the end of the period study (if authors did it). Dependent and independent variables used in the logistic regression analysis should be reported.
Response:The statistical analysis section has been updated thoroughly in the revised manuscript, and relevant details of the dependent and independent variables have been added as suggested. Kindly refer to lines 211-228.
4) Results: results are not clearly reported.
Response:
The result section has been updated thoroughly, and corrections have been made.
5) Discussion: some paragraphs are not related to results.
Response:
Thank you for your comments. The discussion section has been revised thoroughly,and all the irrelevant discussion paragraphs have been deleted.
6) References: some references on the item the authors want to discuss are missing.
Response:
Thank you so much for your kind response. The reference section has been updated with all the new references for the points discussed.
Round 2
Reviewer 3 Report
Authors have done a very good job and the paper has greatly improved. I have only a few minor remarks to highlight:
Please clarify the following sentences
1) lines 29-30: "with a glomerular filtration rate of eGFR<30 and"
2) line 52 I would change "failure" with "disease"
3) lines 58-59: "Women and girls had a higher global majority of kidney failure than men and boys (9.5%) (7.3%)". Besides I would add a reference
4) line 108 I would change "receiving" with "receive"
5) line 125: "In Pakistan, kidney failure is the most common cause of morbidity and mortality.". Besides I would add a reference
6) lines 126-127: According to the National Kidney Federation Registry, 5935 patients were admitted to different dialysis units across the country[32]". When did it happen?
7) lines 157-163: "The multicenter study approach and guidelines provided by the study center, using the Cockcroft-Gault formula gives better risk stratification in cardiovascular events[34]. According to hospital guidelines CKD-EPI was used to estimate the eGFR; however, to reduce factors like weight and body mass index. Cockcroft-Gault formulae have been used to determine the eGFR level of the participants". I cannot understand the meaning of the sentence.
8) lines 187-188: "Assess the pre-study serum potassium, calcium and phosphorus, hemoglobin, and urea levels".
9) lines 198-200: "The KDQOL-Physical SF-36's Health Composite Summary (PCS), Mental Health Composite Summary (MCS), and Kidney Disease Composite Summary (KDCS) domains".
10) lines 212-214: "The following is how HAM-D-U scores are interpreted: Not depressed at all: Mild (subthreshold): 8–13, Moderate (mild): 14–18, Severe (moderate): 19–22, and >23 Very severe (severe)".
11) page 6: please add legend to the figure.
12) page 7: the two figures show the same thing, moreover please complete the legend.
Author Response
1) lines 29-30: "with a glomerular filtration rate of eGFR<30 and"
Response:
Thank you for the comment. The said sentence was updated, and a revised sentence was added in lines 28-29.
2) In line 52, I would change "failure" with "disease."
Response:
Thank you for your comment. The said sentence is updated accordingly in lines 51-52.
3) lines 58-59: "Women and girls had a higher global majority of kidney failure than men and boys (9.5%) (7.3%)". Besides, I would add a reference
Response:
Thank you for highlighting the concern. For the said statement, a reference is added at the end of the statement.
4) line 108 I would change "receiving" with "receive"
Response:
Thank you for the comment. The amendment is being made in the said sentence. Line 100.
5) line 125: "In Pakistan, kidney failure is the most common cause of morbidity and mortality.". Besides, I would add a reference
Response:
Thank you for the comment. The reference was added for the statement. Check lines 114-115.
6) lines 126-127: According to the National Kidney Federation Registry, 5935 patients were admitted to different dialysis units across the country[32]". When did it happen?
Response:
Thank you for highlighting the concern. The statement was updated thoroughly, and the survey year was added to the updated draft.
7) lines 157-163: "The multicenter study approach and guidelines provided by the study center, using the Cockcroft-Gault formula gives better risk stratification in cardiovascular events[34]. According to hospital guidelines CKD-EPI was used to estimate the eGFR; however, to reduce factors like weight and body mass index. Cockcroft-Gault formulae have been used to determine the eGFR level of the participants". I cannot understand the meaning of the sentence.
Response:
The said statement was thoroughly revised for more clarity. Kindly refer to lines 139-147.
8) lines 187-188: "Assess the pre-study serum potassium, calcium and phosphorus, hemoglobin, and urea levels".
Response:
Thank you for the comment. The said sentence is restructured for more clarity. Kindly refer to Lines 162-165.
9) lines 198-200: "The KDQOL-Physical SF-36's Health Composite Summary (PCS), Mental Health Composite Summary (MCS), and Kidney Disease Composite Summary (KDCS) domains".
Response:
Thank you for highlighting the concern. The error is due to sentence restructuring, which is now rectified in updated drafting. Kindly refer to lines 173-176.
10) lines 212-214: "The following is how HAM-D-U scores are interpreted: Not depressed at all: Mild (subthreshold): 8–13, Moderate (mild): 14–18, Severe (moderate): 19–22, and >23 Very severe (severe)".
Response:
Thank you for highlighting this valuable concern. The statement was thoroughly updated, and a more explicit statement was added to the manuscript. Kindly refer to lines 183-194.
11) page 6: please add legend to the figure.
Response:
Thank you for highlighting the mistake. The legend for the said figure is added in the updated draft.
12) page 7: the two figures show the same thing, moreover please complete the legend.
Response:
Thank you for highlighting the concern. The two pictures were there because of the track-changes feature. In the updated drafting, only one figure and a legend were updated accordingly.
Finally, I wonder if authors could correct some grammar mistakes along the text.
Response:
The expert thoroughly proofread the manuscript; few grammatical errors were found and rectified throughout the drafting.